# A numerical framework for simulating the atmospheric variability of supermicron marine biogenic INPs

Isabelle Steinke[1], Paul J. DeMott[2], Grant Deane[3], Thomas C. J. Hill[2], Mathew Maltrud[4], Aishwarya Raman[1], Susannah M. Burrows[1]

[1]Atmospheric Sciences & Global Change, Pacific Northwest National Laboratory, Richland, Washington, USA
[2]Department of Atmospheric Science, Colorado State University, Fort Collins, Colorado, USA
[3]Scripps Institution of Oceanography, University of California at San Diego, La Jolla, California, USA
[4]Climate Ocean Sea Ice Modeling, Los Alamos National Laboratory, Los Alamos, New Mexico, USA

*Correspondence*: Isabelle Steinke (isabelle.steinke@pnnl.gov)

## Abstract

We present a framework for estimating concentrations of episodically elevated high-temperature marine ice nucleating particles (INPs) in the sea surface microlayer and their subsequent emission into the atmospheric boundary layer. These episodic INPs have been observed in multiple ship-based and coastal field campaigns, but the processes controlling their ocean concentrations and transfer to the atmosphere are not yet fully understood. We use a combination of empirical constraints and simulation outputs from an Earth System Model to explore different hypotheses for explaining the variability of INP concentrations, and the occurrence of episodic INPs, in the marine atmosphere. In our calculations, we examine two proposed oceanic sources of high-temperature INPs: heterotrophic bacteria and marine biopolymer aggregates (MBPAs). Furthermore, we assume that the emission of these INPs is determined by the production of supermicron sea spray aerosol formed from jet drops, with an entrainment probability that is described by Poisson statistics. The concentration of jet drops is derived from the number concentration of supermicron sea spray aerosol calculated from model runs. We then derive the resulting number concentrations of marine high-temperature INPs (at 253 K) in the atmospheric boundary layer and compare their variability to atmospheric observations of INP variability. Specifically, we compare against concentrations of episodically occurring high-temperature INPs observed during field campaigns in the Southern Ocean, the Equatorial Pacific, and the North Atlantic. In this case study, we evaluate our framework at 253 K, because reliable observational data at this temperature is available across three different ocean regions, but suitable data is sparse at higher temperatures.

We find that heterotrophic bacteria and MBPAs acting as INPs provide only a partial explanation for the observed high INP concentrations. We note, however, that there are still substantial knowledge gaps, particularly concerning the identity of the oceanic INPs contributing most frequently to episodic high-temperature INPs, their specific ice nucleation activity, and the enrichment of their concentrations during the sea-air transfer process. Therefore, targeted measurements investigating the composition of these marine INPs as well as drivers for their emission are needed, ideally in combination with modeling studies focused on the potential cloud impacts of these high-temperature INPs.

# 1 Introduction

Clouds are important components and drivers of the climate system (Boucher et al., 2013), with mixed-phase cloud processes being a major factor in determining cloud radiative properties (Cesana and Storelvmo, 2017; Vergara-Temprado et al., 2018), the strength of the cloud-climate feedback (McCoy et al., 2016), and the equilibrium climate sensitivity (Tan et al., 2016). The formation of ice crystals in mixed-phase clouds is typically initiated by a small fraction of atmospheric aerosol particles, i.e. ice nucleating particles (Hoose and Möhler, 2012; Murray et al., 2012; Vali et al., 2015; Kanji et al., 2017). While deserts are globally the largest source of ice nucleating particles (INPs) of mineral dust origin, there are areas such as the remote ocean regions where local marine sources dominate the emission of INPs (Burrows et al., 2013). It has been observed that the presence of marine INPs is associated with phytoplankton blooms and microbial degradation of organic material during the decay phase of those blooms (Wang et al., 2015; DeMott et al., 2016; McCluskey et al., 2017a) but the detailed fundamental physical, chemical and biological processes controlling their production and emission to the atmosphere remain unclear (Schiffer et al., 2018).

One group of marine INPs in seawater originates from various types of organic matter in the ocean, with particles often being smaller than 200 nm in diameter (Wilson et al., 2015; Irish et al., 2017). In addition to these small ubiquitous INPs, recent studies have identified another contribution from INPs active at temperatures above 253 K (McCluskey et al., 2018a; Ickes et al., 2020; van Pinxteren et al., 2020), which can be larger and also heat-labile (McCluskey et al., 2018a; Hartmann et al., 2020). The presence of heat-sensitive INPs suggests a contribution from complex organic (e.g. proteinaceous) macromolecules (Christner et al., 2008; Pummer et al., 2015). These high-temperature INPs occur episodically, e.g., after phytoplankton blooms following storm-induced mixing events (Wilbourn et al., 2020) and their identity remains elusive (Ickes et al., 2020).

Aerosolized organic matter in sea spray particles originates from the sea surface microlayer (and possibly the underlying surface layer) which is enriched in surfactants such as carbohydrates, lipids, proteins, and marine biogenic particles such as polymeric aggregates, cells (diatoms, bacteria, viruses), and cell fragments (Garrett, 1967; Patterson et al., 2016; Bertram et al., 2018), with the latter group being the focus of this study. Sea-air transport of organic macromolecules and marine biogenic particles is driven by the bursting of bubbles formed primarily as the result of wave-driven entrainment of air into surface waters. Surface-active organic molecules accumulate in the sea surface microlayer and form films on the surfaces of rising bubbles (Burrows et al., 2014), become aerosolized once the bubbles burst at the air-sea interface, and are subsequently emitted within small film drops (O'Dowd and de Leeuw, 2007). Marine biogenic particles, in contrast, are preferentially emitted through jet drops forming when bubbles collapse and the ejected water jets break up into fragments (Blanchard and Syzdek, 1972; Wu, 2002; Pósfai et al., 2003; Wang et al., 2017). Jet drops are typically the dominant production mechanism for the supermicron sea spray particle population (Wang et al., 2017; Bertram et al., 2018).

Significant advances have been made in developing representations of marine organic INPs appropriate for global climate models (Burrows et al., 2013; Wilson et al., 2015; Huang et al., 2018; McCluskey et al., 2019). However, while modeling studies have demonstrated that INPs from surface-active macromolecules contained in submicron SSA can be predicted skillfully (Huang et al., 2018; McCluskey et al., 2019; Zhao et al., 2021), there are still major challenges in representing episodic contributions from marine, biogenic

particulate INPs larger than 200 nm (Creamean et al., 2019; Trueblood et al., 2020). These INPs are not explicitly described by currently used parameterizations which depend solely on the observed aerosol

surface area or the organic carbon content (Wilson et al., 2015; DeMott et al., 2016; McCluskey et al., 2018a). Recent studies have highlighted that supermicron sea spray particles may contribute significantly to observed marine INP concentrations (Creamean et al., 2019) by acting as carriers for these marine biogenic particles. So far, the sparsity and incompleteness of observational data have posed challenges to developing a process-based source function for these episodically occurring INPs.

There are numerous types of marine particles larger than 200 nm which may accumulate within the sea surface microlayer and act as INPs (McCluskey et al., 2018c), and their identity remains elusive, possibly because most studies have focussed on phytoplankton species and their exudates (Alpert et al., 2011; Ladino et al., 2016; Ickes et al., 2020). The inconclusive findings of these studies point towards the presence of other non-plankton particles which may be only emitted episodically and in connection with

individual events, e.g. related to storm-induced mixing (Wilbourn et al., 2020). These episodic INP emission events are clearly distinct from the background concentration of INPs over the remote oceans contributed by marine organics (McCluskey et al., 2018c). For example, a link between marine ecosystem processes has been hypothesized, as elevated enzymatic activity, higher bacteria concentrations, and increased INP emissions have been concurrently observed during simulated blooms (Wang et al., 2015).

Marine gels form from marine biopolymers, which are excreted by microorganisms and phytoplankton (Verdugo et al., 2008). They are very complex, heterogeneous entities (i.e. biopolymer aggregates) with a potentially variable ice nucleation activity.

In this study, we focus on marine bacteria and marine biopolymer aggregates (MBPAs) as potential sources of episodically emitted marine biogenic INPs. Bacteria and MBPAs accumulate in the sea surface

microlayer (Orellana et al., 2015; Engel et al., 2017; Rahlff et al., 2017) and have been found in aerosol particles emitted from the oceans (Aller et al., 2005; Leck and Bigg, 2008; Orellana et al., 2011). Note that in this study we assign a size of d = 100 nm to MBPAs, with most nanogel particles found in clouds smaller than 200 nm, but with substantial uncertainties regarding their size due to annealing processes, which lead to the formation of microgels that are up to several micrometers in size (Orellana et al., 2011). Therefore,

we have chosen d = 100 nm to represent the order-of-magnitude size of these gel particles. It should also be noted that it is not entirely clear how concentrations of these likely ice-nucleating entities may relate to concentrations of observed INPs.

Even though few studies have investigated the ice nucleation properties of marine bacteria (Fall and Schnell, 1985; Ladino et al., 2016), some bacteria from terrestrial sources are known to be very ice active

(Huang et al., 2021) and ice-nucleating bacteria have been found in isolates derived from coastal air samples (Beall et al., 2021). Note, however, that a quantitative comparison between the ice nucleation efficiencies of marine and terrestrial bacteria remains challenging because only few studies have investigated specifically the immersion freezing properties of marine bacteria. Likewise, ice nucleation efficiencies (e.g. temperature dependent ice-active surface site density values) of MBPAs have not been

measured directly. They could also act as carriers for other potentially ice-active particles (Leck and Bigg, 2008), such as dust resuspended from the ocean surface (Cornwell et al., 2020).

Additionally, there is high uncertainty regarding a potentially selective transfer of marine organic particles across the sea-air interface which may lead to an enrichment of these particles in marine aerosol droplets

relative to the bulk seawater, with enrichment factors (EF) of up to 45 for marine particulate matter (Aller et al., 2005; Rastelli et al., 2017; Gong et al., 2020). EF represents the enhanced concentration of a trace species in the emitted aerosol, relative to its concentration in the seawater. It is typically calculated either relative to the sea salt or sodium mass. EF also depends on the organic/biological species in question as well as the SSA particle size (Quinn et al., 2015). Additionally, EF may also vary between different sampling techniques (Aller et al., 2017).

Here we use a combination of Earth System model simulation outputs and literature data to explore different hypotheses for explaining the variability of INP concentrations, and their episodic occurrence, in the marine atmosphere as part of a novel approach towards quantifying the contribution from episodically elevated high-temperature marine INPs in remote ocean regions. We evaluate to what extent the observed number concentrations and variability of high-temperature INPs can potentially be explained by emissions of sea spray containing either bacteria or marine gel particles, and highlight gaps in our current understanding.

## 2 Analysis of episodic marine INP observations

In this study, we focus on observations of marine INP concentrations from literature data, with the aim to better quantify rare, episodic INP emissions events. INP data was analyzed for three ocean regions where ship-based or coastal campaigns took place: MAGIC, MARCUS, and MHD (see Table 1 for acronym definitions and details). All marine field campaign data analyzed in this study are derived from droplet freezing experiments conducted with the Colorado State University (CSU) Ice Spectrometer (McCluskey et al., 2018b). For these freezing experiments, aerosol particles were collected on polycarbonate membrane filters, frozen for transport, suspended in water and then analyzed with the CSU-Ice Spectrometer. Number concentrations of INPs per liquid volume are evaluated following Vali (1971) and then converted into atmospheric INP concentrations using the known volumes of sampled air and water used to create suspensions from the filter samples (McCluskey et al., 2017b).

*Table 1: Overview of field campaign data used for comparison against offline calculations*

| Campaign | Region | Measurement period | Source |
|---|---|---|---|
| Marine ARM (Atmospheric Radiation Measurement) GPCI (Global Energy and Water Cycle Experiment Cloud System Study Pacific Cross-section Intercomparison) Investigations of Clouds (MAGIC) | Equatorial Pacific | June - September 2013 | DeMott et al., 2016 |
| Measurements of Aerosols, Radiation, and Clouds over the Southern Ocean (MARCUS) | Southern Ocean | November 2017 - March 2018 | Department of Energy Atmospheric Radiation Measurement (ARM) Program Archive |
| Mace Head (MHD) | North Atlantic | August 2015 | McCluskey et al., 2018b |

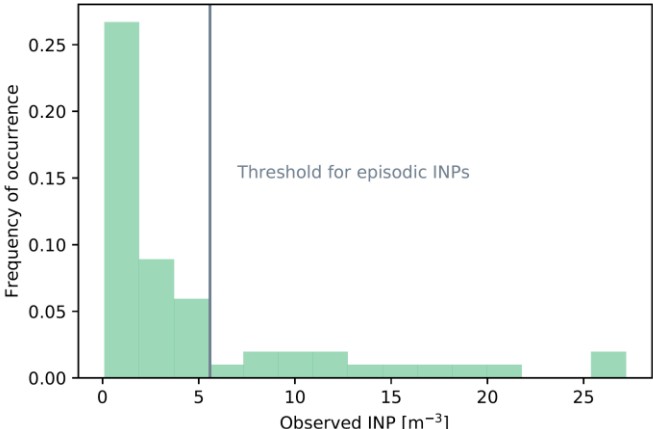

Fig. 1: Combined dataset for three campaigns (Table 1) showing INP concentrations at 253 K; threshold defined as the 75th-percentile for the combined INP dataset (N=57).

Figure 1 shows combined observations of marine INPs at 253 K from the three campaigns. While the majority of observed INP concentrations are well below 5 m$^{-3}$, there are a small number of samples with considerably higher INP concentrations. We define a concentration cutoff for episodic INP events at around 6 m$^{-3}$, which corresponds to the 75th percentile in the distribution of INP concentrations. Note that while the impact from long-range transported and potentially ice-active dust cannot be excluded, it can be

assumed that for very remote regions such as the Southern Ocean the contribution from dust INPs is most likely limited (Burrows et al., 2013). For the MHD measurements, high INP concentrations were observed for marine air masses, with data from air masses with terrestrial influence being excluded from our analysis (McCluskey et al., 2018b). The potential influence of dust INPs is also assumed to be limited for the MAGIC measurements (DeMott et al., 2016). For our analysis, we assume that all episodic INPs derived

from the combined dataset can be attributed to marine INP emissions.

## 3  Numerical framework for estimating marine high-temperature INPs

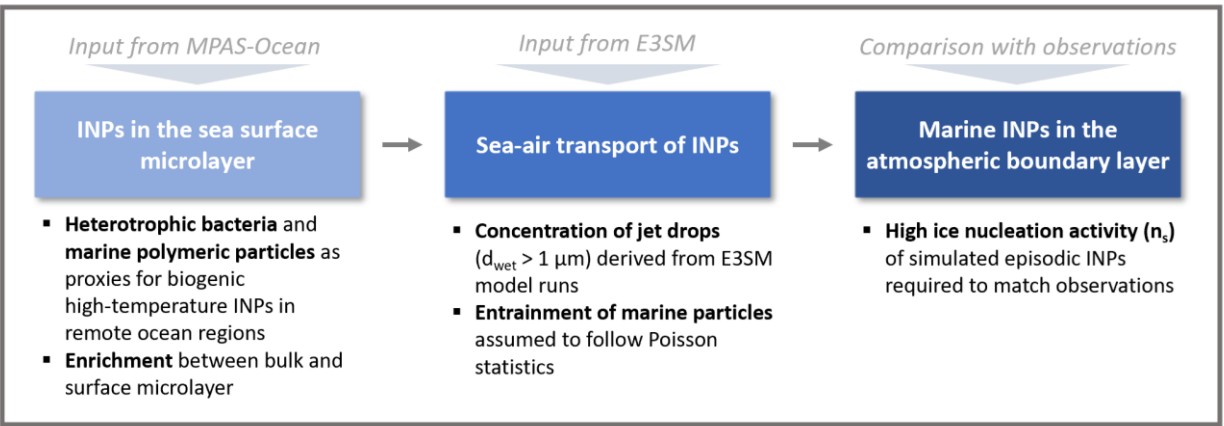

Fig. 2: Schematic overview of numerical framework for estimating concentrations of marine high-temperature

INPs.

Figure 2 shows a schematic overview of the numerical framework that we use to estimate concentrations of episodically emitted high-temperature INPs in remote ocean regions. Our framework accounts for bacteria and MBPA concentrations in the sea surface microlayer and their sea-air transfer through jet drop emission. Based on the abundance of these potentially ice-nucleating particles in the atmospheric boundary layer, we then estimate the range of INP concentrations in the atmospheric boundary, assuming that these marine biogenic particles are mostly transported across the sea-air interface as inclusions in jet drops.

### 3.1 Estimating the abundance of highly ice-active particles in the sea surface microlayer

We use ocean biogeochemistry output from the E3SMv1 MPAS-Ocean component, which relies on the Model for Prediction Across Scales (MPAS) framework (Petersen et al., 2019), to calculate concentrations of bacteria and MBPAs in the sea surface microlayer. Particle concentrations are calculated from monthly mean ocean biogeochemistry output fields for a Rossby Radius of deformation scaling (RRS) mesh with a grid size of RRS30to10km, and then averaged over 10 consecutive years. Further details are described in Brady et al. (2019).

The abundance of heterotrophic bacteria is parameterized as a function of chlorophyll-*a* (chl-*a*) and sea surface temperature (SST). An empirical relationship between these variables was derived by Li et al. (2004), based on 15 years of observations across biogeochemical regimes in all major ocean regions (Li et al., 2004). Based on the chl-*a* concentrations $c_{chl}$, heterotrophic bacteria concentrations $c_{bac}$[cells m$^{-3}$] are given by

$$c_{bac} = 7 \cdot 10^{12} \cdot 10^{0.002 \cdot SST} \cdot c_{chl} \qquad\qquad c_{chl} < 1 \text{ mg m}^{-3} \qquad\qquad (1)$$

$$c_{bac} = 8 \cdot 10^{12} \qquad\qquad c_{chl} > 1 \text{ mg m}^{-3}$$

For the MBPAs, we assume that polymers account for at least 10 % of dissolved organic carbon (DOC) in surface waters (Chin et al., 1998) and that 20 % of these polymers aggregate into MBPAs (Orellana and Leck, 2015). Additionally, we assume a density of 10 kg m$^{-3}$ (Verdugo et al., 2008) and assume that MBPAs contribute 20 % of the total marine gel volume. Note that marine gel particles typically transition between different states of hydration and therefore their density changes over time.

Figures 3 and 4 show the seasonally averaged seawater concentrations of heterotrophic bacteria and MBPAs, respectively, that result from these assumptions. The reported concentrations refer to the top vertical layer of the ocean model, which is roughly 1.5 m deep in the MPAS-Ocean simulations. Inferred seawater bacteria concentrations agree with the range of observed values, e.g. Trueblood et al. (2020). Similarly, for MBPAs, the inferred concentrations agree with observations, with concentrations of marine polymer gel particles (polymers and colloidal gels smaller than 300 nm) up to $10^{22}$ L$^{-1}$ (Orellana et al., 2011).

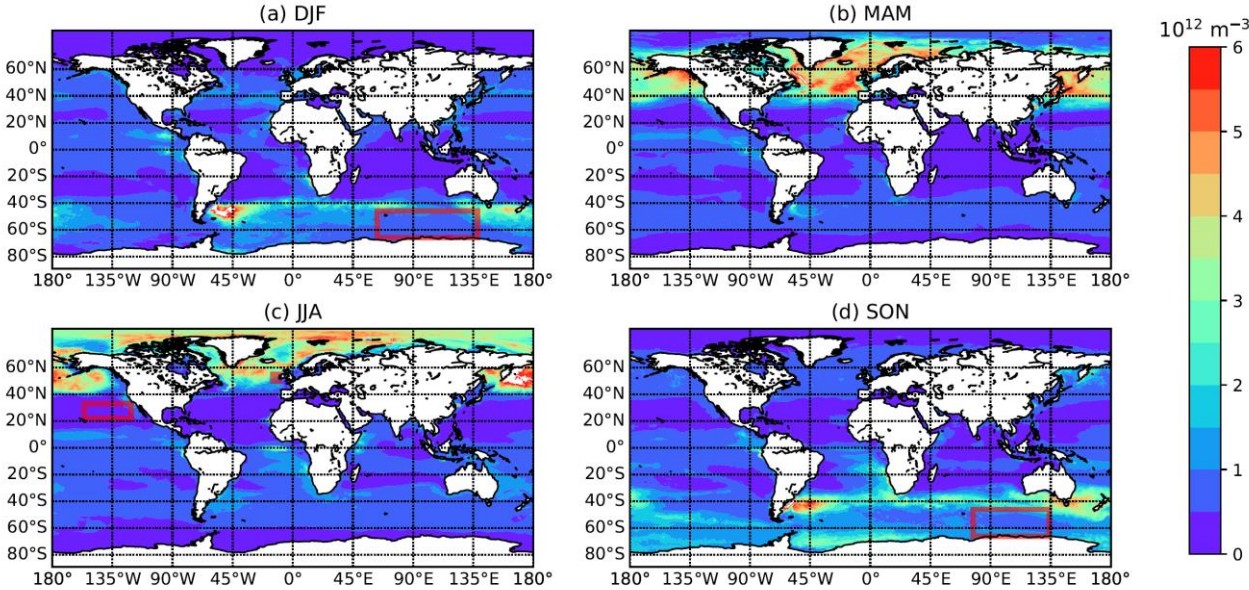

Fig. 3: Global maps of seasonally averaged concentrations of heterotrophic bacteria in bulk seawater as calculated from MPAS-Ocean output (averaged over 10 years), with (a) December - February, (b) March - May, (c) June - August, and (d) September - November; red rectangles indicate areas/seasons for which observations are available (see Table 1).

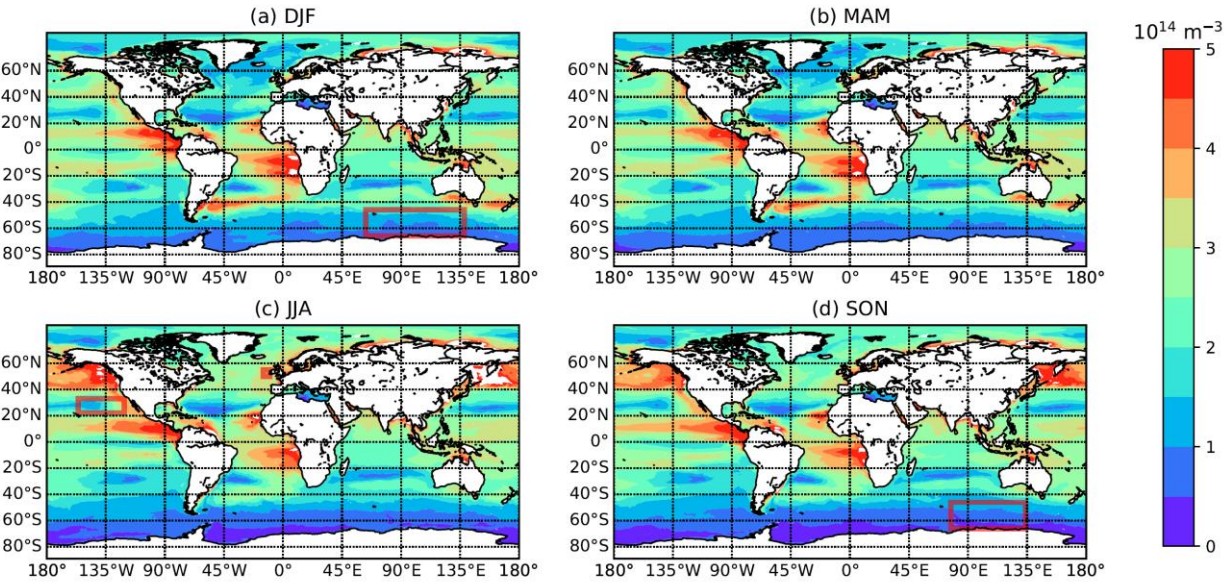

Fig. 4: Global maps of seasonally averaged concentrations of MBPA particles in bulk seawater as calculated from MPAS-Ocean output (averaged over 10 years), with (a) December - February, (b) March - May, (c) June - August, and (d) September - November; red rectangles indicate areas/seasons for which observations are available (see Table 1).

### 3.2 Simulating jet drop concentrations in the marine boundary layer and the entrainment of marine particles into sea spray aerosol

In a second step, we simulate the entrainment (i.e. transfer) of heterotrophic bacteria and MBPAs into supermicron SSA, in order to estimate the number of marine particles (and hence potential INPs) transported to the atmospheric boundary layer (Fig. 2).

Supermicron sea spray aerosol (SSA) particles are a potential carrier of marine biogenic particles larger than 200 nm, e.g. bacteria, intact diatoms, and viruses (Blanchard and Syzdek, 1972; Prather et al., 2013; Patterson et al., 2016). Therefore, we focus on nascent sea spray droplets larger than 1 µm, which are predominantly emitted as jet drops (Blanchard and Syzdek, 1972). Laboratory studies such as Wang et al. (2017) have shown that film drop emission did not significantly contribute to the population of fresh sea spray particles larger than 1 µm. While it is possible for film drop particles to be larger than 1 µm, they are typically smaller than this, and most of the larger sea spray particles are produced through jet drop emissions (de Leeuw et al., 2011). Note, however, that the emission mechanisms of jet and film drops depend strongly on ocean surface processes, which may introduce uncertainties with regard to the size distributions of jet and film drops under atmospheric conditions.

We equate SSA concentrations ($d_{dry} > 0.5$ µm) from monthly averaged simulations with the E3SM Atmosphere Model to concentrations of large jet drops in the atmospheric boundary layer, without parameterizing the jet drop emission process explicitly. Aerosol concentrations in E3SM are simulated using the Modal Aerosol Module (MAM4) and the chemical species included in each mode are illustrated in Liu et al. (2016). The emission of SSA particles is based upon the whitecap parameterization which uses the wind speed at 10 m above the sea surface (Monahan, 1986). SSA particles are produced either directly or indirectly by rain-drop evaporation, resuspension and the major sinks include scavenging by precipitation (interstitial and cloud-borne), wet removal, and dry deposition. Lifetimes of coarse SSA particles (1-10 µm) in E3SM are roughly 0.6 days (Wang et al., 2020). In this study, SSA mixing ratios were simulated by E3SM at $1^o$x$1^o$ resolution with 72 vertical levels. Monthly averaged outputs of surface level SSA number concentrations ($d_{dry} > 0.5$ µm) were used to create maps of jet drop concentrations. Surface maps showing the seasonal variation of jet drop concentrations $c_{jet}[m^{-3}]$ are represented in Fig. 5.

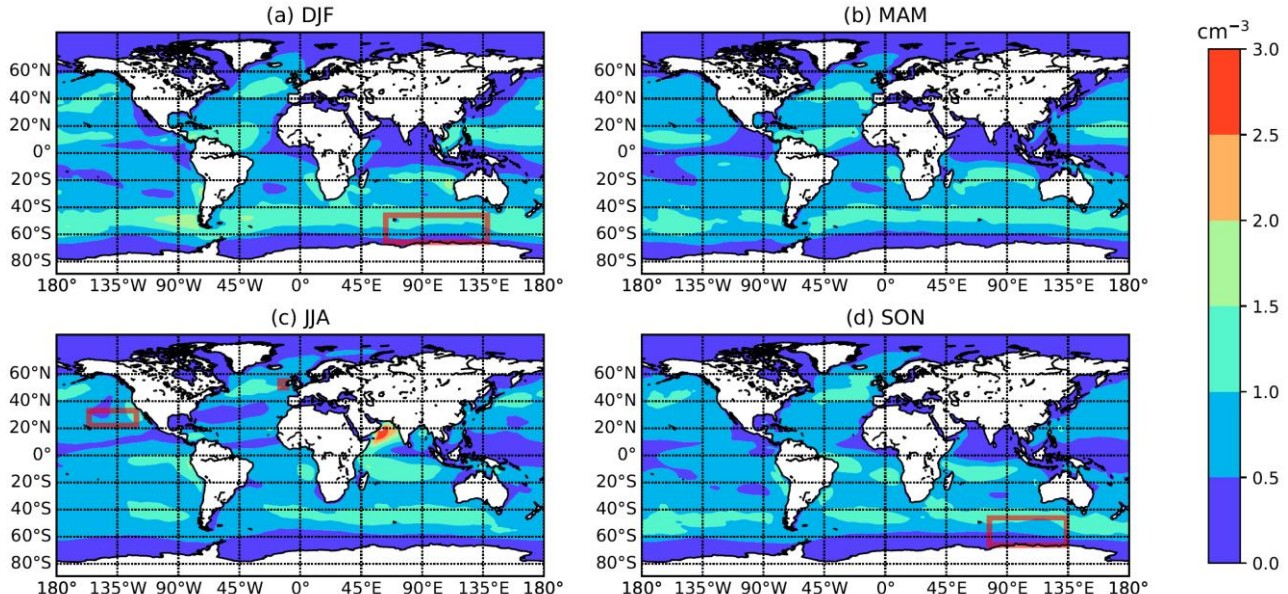

Fig. 5: Global maps of seasonally averaged concentrations of jet drop particles as calculated from E3SM SSA number concentration outputs, with (a) December - February, (b) March - May, (c) June - August, and (d) September - November; red rectangles indicate areas/seasons for which observations are available (see Table 1).


The entrainment of marine particles in supermicron sea spray droplets follows a statistical distribution because of the particles' discrete nature. The probability of finding $k$ marine particles in a drop can be modeled as a Poisson process with an entrainment rate $\lambda$ dependent on the wet drop radius $r_j$[m] ($j$ is the number of bins) and the concentration of marine particles ($c_{mar}$[m$^{-3}$]) in seawater

$$\lambda(r_j) = \frac{4\pi}{3} r_j^3 \cdot c_{mar} \cdot EF \tag{2}$$

EF represents the enrichment of INPs between sea water and sea spray aerosol particles. Note that for our base case (see Table 2), we assume that the processes controlling the transfer of particles from the seawater bulk into a drop are neither selective nor enriching (EF = 1), that each marine particle has the same probability of being entrained in a drop, and that the particles are distributed homogeneously in the
seawater.

### 3.3 Estimating INP concentrations from episodic emissions of marine particles

To compare simulated INP concentrations and observations, we derive estimates for the INP concentrations based on the number of entrained particles (MBPAs and bacteria) which are then aerosolized as inclusions in SSA particles. These entrained particles then get assigned a specific ice nucleation activity (quantified as
the ice nucleation active surface site density, $n_s$).

INP concentrations $c_{INP}$ [m$^{-3}$] are estimated from the average entrainment rate $\lambda$ as

$$c_{INP} = n_s \cdot A_{mar} \cdot \sum_j \lambda(r_j) \cdot c_{jet}(r_j) \tag{3}$$

with $A_{mar}$ being the surface of bacteria, and MBPAs, respectively.

We base our analysis on a comparison between 5 different scenarios as outlined in Table 2. Note that for
our analysis, we assign particle sizes of $d_{MBPA}$ = 100 nm and $d_{bac}$ = 500 nm (Andersen et al., 2016; Orellana

et al., 2011). Assuming a monodisperse size distribution allows us to better demonstrate the impact of different factors investigated in our scenario-based approach which will be described in the following paragraphs. Also, to our knowledge there are currently no observations for size distributions of individual marine species acting as INPs.

For our base case (INP_BASE), we assume that marine particles (i.e. bacteria and MBPAs) are neither enriched nor depleted in the sea surface microlayer, or in the freshly emitted SSA (EF = 1). For 'INP_BASE' we also assume an average ice nucleation active surface site density at 253 K ($n_s = 10^5$ m$^{-2}$). Experimentally measured values of $n_s$ at 253 K have been reported to range from $10^4$ to $10^6$ m$^{-2}$ (DeMott et al., 2016; McCluskey et al., 2018a; Gong et al., 2020), with values for individual marine species expected

to be substantially higher, e.g. $n_s = 10^9$ m$^{-2}$ at 248 K for marine algae (Ickes et al., 2020). Note that the range of observed values is substantially larger than the measurement uncertainties associated with individual values, and therefore we don't consider the impact from measurement uncertainties related to $n_s$ in our study.

For 'INP_BASE', we prescribe the particle concentration in seawater as given by the median of $c_{mar}$ for

each campaign and the corresponding season. We compare our baseline against four cases which are described below:

- 'INP_CONC' investigates the potential impact of high bacteria and MBPA concentrations in seawater, subsequently leading to a higher number of particles being included in supermicron SSA. In our analysis, we define high concentrations as the 90th-percentile of the MBPA and bacteria

concentrations across each campaign domain, respectively.

- Values for EF are highly variable, depending on the species in question, environmental conditions, and sampling techniques. For 'INP_EF' we use EF = 40 which is an upper limit based on the range of EFs found for different marine biogenic particles, e.g. bacteria, viruses, and exopolymers (Aller et al., 2005; Rahlff et al., 2017; Rastelli et al., 2017).

- There is substantial uncertainty regarding the ice nucleation activity of individual marine species because most studies have only looked at the average ice nucleation activity of ambient or simulated SSA particles which are complex mixtures of sea salt, organic surfactants, and marine biogenic particles. Recent studies, however, have found indications that individual species included in supermicron SSA might be characterized by $n_s$ values that are substantially higher (Ickes et al.,

2020; Mitts et al., 2021) than previously observed for SSA. Therefore, for 'INP_ACT' we prescribe a high $n_s$ value with $n_s = 5 \cdot 10^7$ m$^{-2}$, which roughly corresponds to the ice nucleation activity of two phytoplankton species characterized by Ickes et al. (2020). Note that both $n_s$ values are assumed to be representative of the observed ice nucleation efficiencies at 253 K.

- 'INP_MAX' is the upper limit of simulated INP concentrations, based on high estimates of EF, $n_s$,

and the concentration of bacteria and MBPAs in seawater.


*Table 2: Overview of parameters for simulated INP concentrations*

| Scenarios | Parameters |
|---|---|
| INP_BASE | EF = 1, $n_s = 10^5$ m$^{-2}$, median concentration of marine particles |
| INP_CONC | EF = 1, $n_s = 10^5$ m$^{-2}$, high concentration of marine particles |
| INP_EF | EF = 40, $n_s = 10^5$ m$^{-2}$, median concentration of marine particles |
| INP_ACT | EF = 1, $n_s = 5 \cdot 10^7$ m$^{-2}$, median concentration of marine particles |
| INP_MAX | EF = 40, $n_s = 5 \cdot 10^7$ m$^{-2}$, high concentration of marine particles |


## 4 Comparison with observations of episodic marine INP emissions

Figures 6-9 show results from simulated INP concentrations compared to observed episodic INP concentrations (see cases described in Table 2). The violin plots represent the distribution of INP values for each season and across the geographic area where each campaign has taken place. In each figure, we show
INP concentrations observed at 253 K (left panel, with representative error bars for the lowest INP concentrations), with episodic events characterized by concentrations above the threshold derived from the combined INP dataset (Fig. 1). Note that the number of available data points varies between the three campaigns, with the episodic nature of INP emissions most pronounced for the MARCUS dataset (Figs. 6 and 7).

Observed INP concentrations are compared against simulated INP concentrations, using different scenarios (Table 2) to quantify the potential impact of varying $n_s$, EF, and the concentration of marine biogenic particles in seawater.

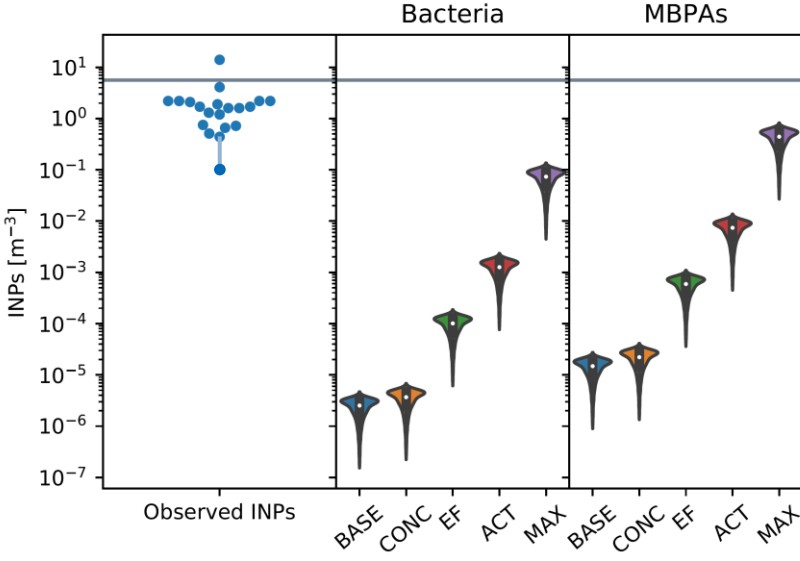

Fig. 6: Left panel: each point corresponds to a single filter measurement with the value of the INP concentration
indicated by their location on the y-axis; the points are arranged in the approximate shape of a as a visual cue to facilitate the comparison with the simulated values; middle and right panels: simulated INP concentrations from bacteria (middle) and MBPAs (right) at 253 K, with violinplots representing the distribution of values across the

relevant spatial domain, and inner box plots indicating the median and interquartile values; the grey line indicates the threshold above which emissions are considered as episodic events; parameters for scenarios as listed in Table 2 (campaign: MARCUS, DJF).

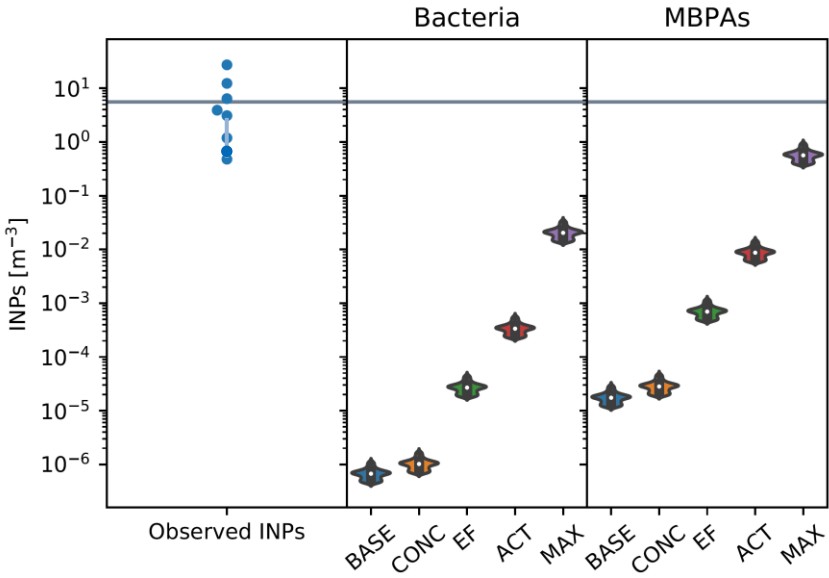

Fig. 7: Left panel: each point corresponds to a single filter measurement with the value of the INP concentration indicated by their location on the y-axis; the points are arranged in the approximate shape of a as a visual cue to facilitate the comparison with the simulated values; middle and right panels: simulated INP concentrations from bacteria (middle) and MBPAs (right) at 253 K, with violinplots representing the distribution of values across the relevant spatial domain, and inner box plots indicating the median and interquartile values; the grey line indicates the threshold above which emissions are considered as episodic events; parameters for scenarios as listed in Table 2 (campaign: MARCUS, SON).

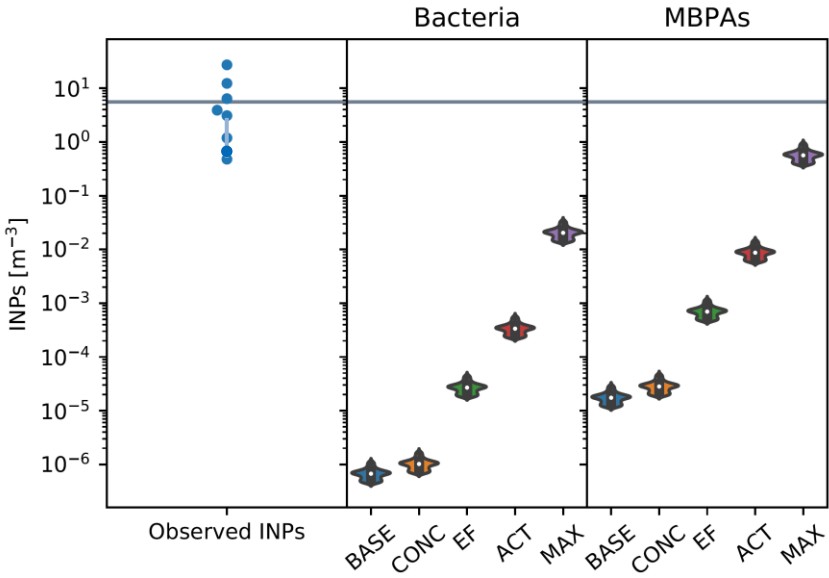

Fig. 8: Left panel: each point corresponds to a single filter measurement with the value of the INP concentration indicated by their location on the y-axis; the points are arranged in the approximate shape of a as a visual cue to

facilitate the comparison with the simulated values; middle and right panels: simulated INP concentrations from bacteria (middle) and MBPAs (right) at 253 K, with violinplots representing the distribution of values across the relevant spatial domain, and inner box plots indicating the median and interquartile values; the grey line indicates the threshold above which emissions are considered as episodic events; parameters for scenarios as listed in Table 2 (campaign: MAGIC, JJA).

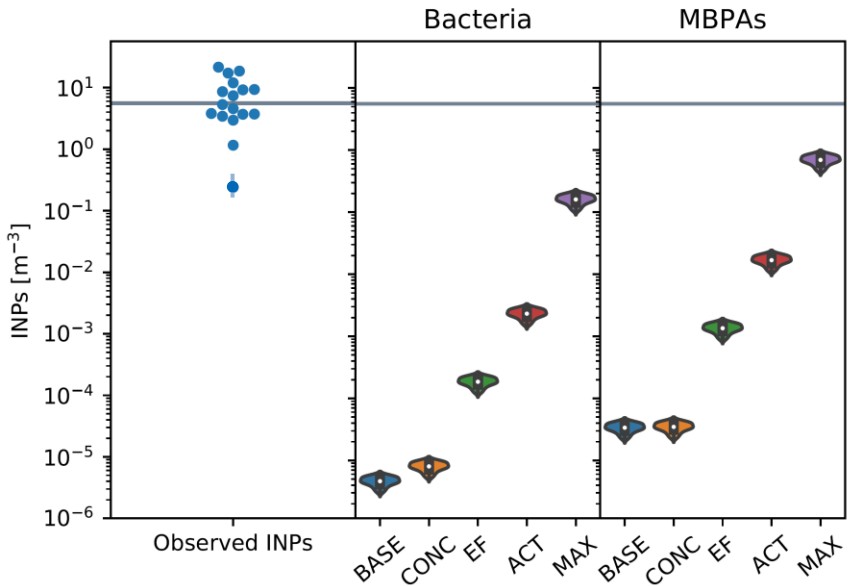

Fig. 9: Left panel: each point corresponds to a single filter measurement with the value of the INP concentration indicated by their location on the y-axis; the points are arranged in the approximate shape of a as a visual cue to facilitate the comparison with the simulated values; middle and right panels: simulated INP concentrations from bacteria (middle) and MBPAs (right) at 253 K, with violinplots representing the distribution of values across the relevant spatial domain, and inner box plots indicating the median and interquartile values; the grey line indicates the threshold above which emissions are considered as episodic events; parameters for scenarios as listed in Table 2 (campaign: MHD, JJA).

Our results show that compared to the variation in INP concentrations in sea water, the enrichment factor as well as the particle type dependent ice nucleation activity have a larger impact on the estimated jet drop INP concentrations. The relative variability in simulated INP concentrations – which is driven by the variability in jet drop concentrations across each domain – is largest for the two MARCUS cases. For all three campaigns, we observe a significant underprediction of INP concentrations by at least one order of magnitude. This finding indicates that contributions from larger, episodically occurring INPs might be limited to very specific conditions (e.g. ocean ecosystems, seasons, wind speed, and SST) or that there might be other processes leading to an enrichment of INPs associated with marine biogenic particles, as enrichment factors of ~ 200 have been observed in recent laboratory experiments (private communication with T. Hill). For the MBPAs, an enrichment factor of 200 would lead to a close agreement between observed and simulated INP concentrations (using the maximum assumptions). Also, the relative contribution of film and jet drops to the emission of SSA is virtually unknown under atmospheric conditions, and hence also the number concentration of jet drop particles and their size distribution. Note

that simulated film drop INP concentrations based on an average ice nucleation activity at 253 K of $n_s=10^5$ m$^{-2}$ (McCluskey et al., 2018c) generally range between $10^{-4}$ and $10^{-2}$ m$^{-3}$ (data not shown) which is substantially lower than the observed INP concentrations. Therefore, model simulations may not fully capture the contribution of jet drop emission to the overall SSA population. Additionally, it should be noted

that the factors that we varied for our sensitivity study are subject to substantial uncertainties and therefore additional studies are warranted.

In particular, we identify the following gaps based on our analysis:
- To further improve our process-level understanding of factors driving the occurrence of episodic INPs (e.g. marine biological activity), targeted observations of episodic INP emissions should be

conducted, focusing on the frequency of these rare events and a process-level understanding of the sea-air transport of relevant marine INPs (larger than 200 nm). A more detailed understanding of the transport mechanisms is also a pre-requisite for better quantifying enrichment factors under different conditions and for different particle species.
- Another key uncertainty which could be addressed by laboratory studies is the temperature-

dependent ice nucleation efficiency (e.g. as $n_s$ values) of individual marine species that contribute directly or indirectly to the observed INP population, particularly under mixed-phase conditions. Measurements of the ice nucleation efficiencies would also help to identify individual particle species which are most likely to contribute to elevated INP concentrations during episodic INP emission events.


## 5 Conclusions and outlook
In this study, we present a novel framework for exploring hypotheses regarding the emissions of marine, high-temperature INPs. This framework is used to simulate hypothetical sources of INPs in seawater (bacteria and MBPAs) as well as their transport to the marine atmospheric boundary layer via sea spray jet

drop production. We also highlight the gaps that currently prevent the development of a source function for these INPs and which are mostly related to a lack of data to fully inform and constrain such a function.

Better quantifying the potential climate impacts from episodic INP emissions over the remote ocean requires a model-based assessment of their cloud impacts. More studies using cloud-resolving models (with appropriate cloud microphysics) are needed to evaluate the impacts of episodic increases in INPs on marine

cloud processes. These studies can provide guidance on how accurately we need to be able to characterize the INP activities of marine biogenic particles, in order to predict cloud impacts associated with these episodically emitted marine particles.

Targeted observations characterizing these episodic INPs would be needed to develop a more detailed process-level understanding of their occurrence. In particular, measurements are needed which comprise not

only quantification of the number concentration of INPs at a certain temperature but also measurements of the particle size distribution (including the range up to 5-10 μm), as well as single-particle analyses (e.g., using microscopy methods) that can provide insight into their chemical, physical and biological identities. These measurements can serve as a starting point for identifying relevant INP types and for developing tailored parameterizations that can be tested in models, ranging from large-eddy simulations to large-scale

climate models. Note, however, that the collection of a comprehensive dataset allowing for the development of a globally representative source function for episodically occurring INPs in marine environments would

be extremely challenging due to the complexity and variability of ocean ecosystems. Therefore, more modeling studies are needed to quantify the sensitivities of marine mixed-phase clouds to the presence of these episodic high-temperature INPs and to develop a targeted approach for future measurements.


### Code availability

All E3SM model codes (doi:10.11578/E3SM/dc.20180418.36) may be accessed through GitHub (https://github.com/E3SM-Project/E3SM/releases/tag/v1.0.0).


### Data availability

Observational data used in this publication is available through the data archive of the Atmospheric Radiation Measurement (ARM) User Facility, a U.S. Department of Energy (DOE) Office of Science user facility managed by the Office of Biological and Environmental Research.


### Author contributions

IS and SB designed the study. AR and MM provided output from E3SM and MPAS-Ocean simulations. IS conceptualized the numerical framework presented in this study, with contributions from SB, PD, GD and TH. IS conducted all analyses and prepared the manuscript, with all co-authors contributing to reviewing

and editing the draft.

### Competing interests

The authors declare that they have no conflict of interest.

**Acknowledgements**

This research was supported by the U.S. Department of Energy (DOE), Office of Science, Office of Biological and Environmental Research through the Early Career Research Program and used data from the Atmospheric Radiation Measurement Climate Research Facility, a DOE Office of Science User Facility. A portion of the research for this study was performed using resources available through Research Computing

at Pacific Northwest National Laboratory (PNNL). E3SM simulation data were obtained from the Energy Exascale Earth System Model project, sponsored by the U.S. Department of Energy, Office of Science, Office of Biological and Environmental Research. PD and TH acknowledge support from ASR under grant No. DE-SC0018929 and grant No. DE-SC0021116. GBD acknowledges funding by the National Science Foundation through the NSF Center for Aerosol Impacts on Chemistry of the Environment (CAICE), a

Center for Chemical Innovation (CHE-1801971).

The Pacific Northwest National Laboratory is operated for DOE by Battelle Memorial Institute under contract DE-AC05-76RL01830.

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
