# Peer review of "A numerical framework for simulating the atmospheric variability of supermicron marine biogenic INPs"

_Atmospheric Chemistry and Physics, 2021_

## Referee Comment (RC1)

This manuscript presents a novel framework for estimating concentrations of episodically elevated high-temperature marine ice-nucleating particles (INPs) in the ocean surface and their subsequent emission into the ambient. Although there are larger uncertainties of this method and several scientific gaps (such as (1) size-resolved measurement of INPs, marine bacteria and MBPAs, (2) enrichment factors of marine bacteria and MBPAs, and (3) ice nucleation surface density site of bacteria and MBPAs in different temperatures) remain unclear, this work still gives us a new way to link the ocean activities and INPs contribution. The following major comments must be satisfactorily addressed before consideration for publication.

Major comments:

1. In this study, the production of bacteria and MABPs is assumed by jet drops. However, considering the size of MBPAs (100 nm), the production of MBPAs via film drops might be also important, as the small organics are enriched in the sea surface microlayer. Besides, can bacteria be produced via film drops? In terms of the number concentration, particles produced via film drops are much higher than jet drops.

2. Will the diameter of MBPAs and bacteria affect your model results? And How? I preassume that with a precious number size distribution of MBPAs and bacteria, the model results will be more close to reality.

3. I think it needs more discussion concerning the model results (Fig. 6-9). For example, (1) the EF and ns linearly correlated with model INP number concentration, but the high concentration (90th) of bacteria and MABPs are not. (2) Which factor is more important to control the model results? It seems the high bacteria and MBPAs concentration is less important than EF and ns numbers. (3) In the MARCUS site, field INP number concentration in DJF is higher than SON, the model results also follow the same trend. (4) Why model results in the MARCUS show larger variation, but in the MAGIC and MHD show small variation? But the field results in all sites show comparable variation.

4. A suggestion: It would be nice to show the simulated INP concentration from bacteria and MBPAs on a global map and compare it to previous studies, such as Burrow et al., 2013.

5. Why you only compare the episodic INPs, i.e., the highest 75$^{th}$ percentile INPs? If you compare the median of all INP number concentrations with your model, the results might be comparable. As the paper is well structured, I just need clarification in the response.

Minor comments:

1. Lines 99-102: I am not clear why you have this paragraph in the abstract. The logic flow seems been interrupted by this paragraph. Please clarify it.

2. Lines 116-120: It worth mentioning that the enrichment factor is also related to particle sizes. See Fig. 5 in Quinn et al., 2015.

3. In Fig. 6-9, Error bars of field measurements (blue dots) are needed. I also suggest combining Fig. 6-9 into one figure, with (a) to (d) panels.

References:

Burrows S M, Hoose C, Pöschl U, et al. Ice nuclei in marine air: Biogenic particles or dust?[J]. Atmospheric Chemistry and Physics, 2013, 13(1): 245-267.

Quinn P K, Collins D B, Grassian V H, et al. Chemistry and related properties of freshly emitted sea spray aerosol[J]. Chemical reviews, 2015, 115(10): 4383-4399.

---

## Author Comment (AC1)

**Response to reviewers: "A numerical framework for simulating episodic emissions of high temperature marine INPs" by Steinke et al.**

We would like to thank both reviewers for thoroughly reviewing our study and providing helpful feedback. In the following responses, we will reply to each comments point-by-point, with our replies highlighted in green and changes to the manuscript highlighted in blue.

**Reviewer #1**

In the paper "A numerical framework for simulating episodic emissions of high temperature marine INPs" by Steinke et al. 2020 the authors present a framework based on numerical simulations and measurements (field and laboratory) to estimate the role of marine biopolymer aggregates and marine bacteria for episodic emissions of INP at 253 K. The combination of different tools is well done and the results reveal interesting aspects when it comes to marine INP. Therefore the study is suitable for ACP after some minor revisions.

**General comments:**

−   One aspect of the paper, which is a bit unclear to me as a reader is the focus on 253 K as the one and only temperature for the analysis. Why was the analysis not done for several temperatures? Why was the INP concentration calculated with this constant temperature? Are the measurements limited to 253 K? How different would the analysis look like in a colder/warmer temperature regime, e.g. Fig. 1?
    Related to that: 253 K is not really "high-temperature" in the context of immersion freezing in mixed-phase clouds (one would assume rather 258 or 263 K as "high temperature"). I would suggest to adapt the title (see also specific comments) and the text according to that.
    We have chosen to focus our analysis on 253 K, because this a temperature range where contributions from biological particles acting as INPs are still relevant for the remote oceans, and at the same time, the statistics of the droplet freezing measurements are good enough to analyze the long tail of the INP concentration distribution. Also, we needed to have reliable data across different campaigns to compare between different oceanic regions, which further motivated our choice to analyze results at 253 K. We believe that our framework can be applied at temperatures warmer than 253 K, but more observational data would be needed to extend our analysis. We have, however, changed the title accordingly (see response to specific comments).
−   There are many assumptions done in the course of the paper/framework. This is necessary, but unfortunately often mentioned without reflections on the uncertainty etc. The paper would gain a lot more value if a reflection and if possible also a discussion for related uncertainties could be added. Sometimes also the legitimizations of assumptions are missing or not described in detail.
    We have added some more information regarding our assumptions throughout the manuscript and hope that our parameter choices are now clearer – please see our responses to the specific comments raised by reviewer #1.

For example: it would be good to discuss the limitation that comes with assuming that all episodic INP are coming from marine emissions (section 2). What is the legitimation in doing so and how much would your result change assuming other potential episodic sources?

In our study we explore the potential role of marine biogenic particles in explaining observed episodic INP emission events. However, there are indeed other INP sources (e.g. dust) which may also contribute to episodically-high INP emissions, particularly close to landmasses. We have changed the title accordingly (see response to the specific comments raised by reviewer #1).

Another example is the amount of measurements when comparing the results (Fig. 3-5, 6-9) - is it sufficient, where are observations missing etc. (when analysing the variability of the numerical simulations...).

We have expanded the last paragraph of section 4 by including more details regarding measurements that would be needed to further constrain the parameters in our framework.

**Specific comments:**

— The title could be more concise. "High-temperature INP" is not clearly defined and many readers might associate that with higher temperatures than 253 K.

We have decided to change the title to "A numerical framework for simulating the atmospheric variability of supermicron marine biogenic INPs" to better reflect the content of our manuscript. Additionally, we added a sentence to the abstract (l. 27ff):

In this case study, we evaluate our framework at 253 K, because reliable observational data at this temperature is available across three different ocean regions, but suitable data is sparse at higher temperatures.

— Page 2, line 49: Are deserts really the largest source when also considering biological INP?

In the manuscript we are referring to deserts as a major source of mineral dust INPs. However, we don't necessarily imply that this statement is also valid for the total concentration of atmospheric INPs. Biological particles may contribute significantly to the population of atmospheric INPs, particularly at small supercoolings.

— Page 3, line 93: It would be helpful to add another 1-2 sentences of explanation for your main hypothesis why these studies point towards episodically emission.

We have expanded our hypothesis (l. 96ff):

The inconclusive findings of these studies point towards the presence of other non-plankton particles which may be only emitted episodically and in connection with individual events, e.g. related to storm-induced mixing (Wilbourn et al., 2020). These episodic INP emission events are clearly distinct from the background concentration of INPs over the remote oceans contributed by marine organics (McCluskey et al., 2018c).

— Page 3, line 107-108: Orellana et al. found gel particles in clouds to be smaller than 100nm, but you assign 100 nm in study - isn't that a slight contradiction?

We have rephrased this sentence to make it more consistent:

Note that in this study we assign a size of d = 100 nm to MBPAs, with most nanogel particles found in clouds smaller than 200 nm, but with substantial uncertainties regarding their size due to annealing processes, which lead to the formation of microgels which are up to several micrometers in size (Orellana et al., 2011). Therefore, we have chosen d = 100 nm to represent the order-of-magnitude size of these gel particles.

– Page 3, line 110: Are the marine bacteria in the mentioned studies similar ice-active as bacteria from terrestrial sources? Be more specific here.

At this time, a quantitative comparison between the ice nucleation efficiencies of bacteria emitted by terrestrial and marine sources is not possible, because previous studies characterizing the INP activity of marine bacteria have investigated only the deposition nucleation properties of marine bacteria (Ladino et al., 2016) or did not derive surface dependent metrics from their observations (e.g., Fall and Schnell, 1985). However, recently a study showed that a third of bacteria and fungi from isolates collected from coastal air samples (Southern California) were able to initiate ice formation (Beall et al., 2021).

We have added one more sentence and a reference to the study by Beall et al. (2021) in l. 118ff: Note, however, that a quantitative comparison between the ice nucleation efficiencies of marine and terrestrial bacteria remains challenging because only few studies have investigated specifically the immersion freezing properties of marine bacteria.

– Page 4, line 132-133: I assume the INP are collected on filter and then investigated in droplet freezing experiments? Add this information as well to be unambiguous.

This information has been added (l. 157 f): For these freezing experiments, aerosol particles were collected on polycarbonate membrane filters, frozen for transport, suspended in water and then analyzed with the CSU-Ice Spectrometer.

– Fig 1: How is the INP concentration derived from the measurements for Fig. 1? Is the frozen fraction from the droplet freezing experiments estimated at 253 K and then multiplied with the ambient aerosol concentration present at the field measurements (at which height/temperature was these measured/is that consistent)?

For the droplet freezing experiments, number concentrations of INPs per volume of liquid are derived as $-\ln(f)/V_a$ (Vali, 1971), with f the number of frozen aliquots in the PCR trays used in the droplet freezing experiments and $V_a$ the aliquot liquid volume. These INP concentrations per volume of liquid are then converted into atmospheric INP concentrations using the known volumes of sampled air and water used to create suspensions from the filter samples. We have added this information in l. 158 ff: Number concentrations of INPs per liquid volume are evaluated following Vali (1971) and then converted into atmospheric INP concentrations using the known volumes of sampled air and water used to create suspensions from the filter samples (McCluskey et al., 2017).

– Page 7, line 212: How crude is it to assume that SSA concentrations = concentration of large jet droplets? What is the uncertainty related to this assumption?

For the purposes of this analysis, we have assumed that supermicron SSA concentrations are equivalent to jet drop concentrations. Laboratory studies such as Wang et al. (2017) have shown that film drop emission did not significantly contribute to the population of fresh sea spray particles larger than 1 μm. While it is possible for film drop particles to be larger than 1 μm, they are typically smaller than this, and most of the larger sea spray particles are produced through jet drop emissions (de Leeuw et al., 2011). Note, however, that the emission mechanisms of jet and film drops depend strongly on ocean surface processes which may introduce uncertainties with regard to the size distributions of jet and film drops under atmospheric conditions. We have reflected this information now also in the manuscript (l. 241 ff):

Laboratory studies such as (Wang et al., 2017) have shown that film drop emission did not significantly contribute to the population of fresh sea spray particles larger than 1 μm. While it is possible for film drop particles to be larger than 1 μm, they are typically smaller than this, and most of the larger sea spray particles are produced through jet drop emissions (de Leeuw et al., 2011). Note, however, that the emission mechanisms of jet and film drops depends strongly on ocean surface processes, which may introduce uncertainties with regard to the size distributions of jet and film drops under atmospheric conditions.

– Page 7, line 217: Is the lifetime so short because of the large size of the particles?
Yes, coarse sea spray particles are efficiently removed through dry deposition and below-cloud wet removal. However, simulated lifetimes across different GCMs vary substantially, with E3SMv1 possibly overpredicting dry deposition, e.g. in the case of mineral dust (Wu et al., 2020).

– Page 7, line 242: What is the size of the bacteria based on?
We have added a reference.

– Section 3.3: Is n_s calculated based on these two fixed sizes? What is the uncertainty in n_s and INP concentration later assigned to this assumption?
The $n_s$ values used in our study aim to represent a range of reasonable values for observed ice nucleation efficiencies, with experimentally measured values of $n_s$ at 253 K ranging between $10^4$ to $10^6$ $m^{-2}$ (DeMott et al., 2016; McCluskey et al., 2018a; Gong et al., 2020). For individual algal species, however, higher $n_s$ values are expected, e.g. $n_s = 10^9$ $m^{-2}$ at 248 K (Ickes et al., 2020). Based on these reported values, we use $n_s = 10^5$ $m^{-2}$, a value representative of the center of this range, for our BASE case, and $n_s = 5 \cdot 10^7$ $m^{-2}$ for our ACT case. Note that the range of observed $n_s$ values is substantially larger than the measurement uncertainties associated with individual values, and therefore we don't consider the impact from measurement uncertainties related to $n_s$ in our study.
We have now re-phrased this paragraph slightly (l. 293ff):
Experimentally measured values of $n_s$ at 253 K have been reported to range from $10^4$ to $10^6$ $m^{-2}$ (DeMott et al., 2016; McCluskey et al., 2018a; Gong et al., 2020), with values for individual marine species expected to be substantially higher, e.g. $n_s = 10^5$ $m^{-2}$ at 248 K for marine algae (Ickes et al., 2020). Note that the range of observed values is substantially larger than the measurement uncertainties associated with individual values, and therefore we don't consider the impact from measurement uncertainties related to $n_s$ in our study.

– Section 3.3: I am missing a formula here how exactly you transferred c_mar to an INP concentration, please name/explain the procedure in more detail (c_mar leads to lambda leads to ...). Also the units are missing for the specific variables, for example lambda (unit less?).
We have added eq.3 and units.

– Page 9, line 264: At 253 K (n_s)?
We added a sentence in l. 318f:
Note that both $n_s$ values are assumed to be representative of the observed ice nucleation efficiencies at 253 K.

– Fig. 6-9: What does the shape of the calculated points represent (which uncertainties etc.)?
We have added a clarification in l. 328f:
The violin plots represent the distribution of INP values for each season and across the geographic area where each campaign has taken place.
Additionally, we have also expanded the description given in the figure captions for Fig. 6-9.

- Page 12, line 306: Would a factor of 200 be enough?
  We have added the following sentence to the manuscript (l. 390f):
  For the MBPAs, an enrichment factor of 200 would lead to a close agreement between observed and simulated INP concentrations (using the maximum assumptions).
- Section 4 enumeration gaps: the first point does not necessarily follow the presented analysis, sea-air transport could be a separate point instead of being included in the second point, the third point could be more concrete: what to focus on etc..
  We have re-phrased this paragraph to focus on two points which are directly related to our analysis, i.e. the enrichment factors and the ice nucleation activities of marine particles.
  - To further improve our process-level understanding of factors driving the occurrence of episodic INPs (e.g. marine biological activity), targeted observations of episodic INP emissions should be conducted, focusing on the frequency of these rare events and a process-level understanding of the sea-air transport of relevant marine INPs (larger than 200 nm). A more detailed understanding of the transport mechanisms is also a pre-requisite for better quantifying enrichment factors under different conditions and for different particle species.
  - Another key uncertainty which could be addressed by laboratory studies is the temperature-dependent ice nucleation efficiency (e.g. as $n_s$ values) of individual marine species that contribute directly or indirectly to the observed INP population, particularly under mixed-phase conditions. Measurements of the ice nucleation efficiencies would also help to identify individual particle species which are most likely to contribute to elevated INP concentrations during episodic INP emission events.
  Future priorities for modeling and observations will be discussed in the next section:
  More studies using cloud-resolving models (with appropriate cloud microphysics) are needed to evaluate the impacts of episodic increases in INPs on marine cloud processes. These studies can provide guidance on how accurately we need to be able to characterize the INP activities of marine biogenic particles (e.g. the temperature dependence of $n_s$), in order to predict cloud impacts associated with these episodically emitted marine particles.
- It would be great if an hypothesis could be added in the end for the gap seen in the analysis: which species are maybe/probably missing etc.. Which further measurements are needed...
  We have added one more sentence (see previous response) about the importance to quantify the ice nucleation efficiencies of individual species, in order to identify species which might be particularly important for the elevated INP concentrations observed during episodic emission events.

**Technical corrections:**

- Page 2, line 82: Split up in two sentences (from , which...).
  Done.
- Page 3, line 104: "larger" needs a reference.
  We have rephrased this sentence (l. 106ff):
  In this study, we focus on marine bacteria and marine biopolymer aggregates (MBPAs) as potential sources of episodically emitted marine biogenic INPs.
- Page 5, line 169: Space missing (RRS30 to).

We did not change this; here we are following the naming convention for E3SM grid sizes.

- Page 7, line 201: Flip entrainment and jet drop concentration (the second follows the first). Maybe reformulate entrainment to transfer (entrainment reminds of aerosols being entrained in a cloud/air parcel)?

  We have kept the order as-is to maintain the flow because the jet drop concentrations are variables that are derived from our simulations (similarly as the fields for bacteria and MBPA concentrations). However, we have rephrased the first sentence in the aforementioned section to clarify that entrainment in our case means transfer (l. 235):

  In a second step, we simulate the entrainment (i.e. transfer) of heterotrophic bacteria and MBPAs into supermicron SSA, in order to estimate the number of marine particles (and hence potential INPs) transported to the atmospheric boundary layer (Fig. 2).

- Page 8, line 231: Add that EF is the entrainment factor? Is r_c defined earlier?

  We have renamed $r_c$ as $r_j$, with j being the number of bins. We have also added a clarification regarding EF (the enrichment factor) in l. 270:

  EF represents the enrichment of INPs between sea water and sea spray aerosol particles.

  We have also added a definition of enrichment factors in l. 140ff:

  EF represents the enhanced concentration of a trace species in the emitted aerosol, relative to its concentration in the seawater. It is typically calculated either relative to the sea salt or sodium mass.

- Add . at the end of the figure captions (inconsistent in the current version of the manuscript).

  Done.

**Reviewer #2**

This manuscript presents a novel framework for estimating concentrations of episodically elevated high-temperature marine ice-nucleating particles (INPs) in the ocean surface and their subsequent emission into the ambient. Although there are larger uncertainties of this method and several scientific gaps (such as (1) size-resolved measurement of INPs, marine bacteria and MBPAs, (2) enrichment factors of marine bacteria and MBPAs, and (3) ice nucleation surface density site of bacteria and MBPAs in different temperatures) remain unclear, this work still gives us a new way to link the ocean activities and INPs contribution. The following major comments must be satisfactorily addressed before consideration for publication.

**Major comments:**

1. In this study, the production of bacteria and MABPs is assumed by jet drops. However, considering the size of MBPAs (100 nm), the production of MBPAs via film drops might be also important, as the small organics are enriched in the sea surface microlayer. Besides, can bacteria be produced via film drops? In terms of the number concentration, particles produced via film drops are much higher than jet drops.

Since most of the surface area and mass of material in sea spray particles are produced through jet drop emissions (jet drops emerge through the base of the bubble upon its rupture; this depression is enriched with SML material that is swept up with the jet drop), the contribution by film drops is minor. The bubble film is thin, ranging from 0.5 to 2 µm in thickness in bubbles that persist for up to 10 s (Modini et

al., 2013), so bacteria may be embedded within it, but they are more likely to be present within the jet drops. We are therefore slightly underestimating marine INP concentrations by neglecting the contribution of film drops which dominate the INP background over remote ocean regions. However, the observed episodic INP emissions are characterized by INP concentrations significantly exceeding average background INP concentrations (Fig. 1). With these background INPs (i.e. marine organics) being moderately ice-active at 253 K, this contribution can be assumed to be comparatively small in the case of our episodic INP emission events.

2. Will the diameter of MBPAs and bacteria affect your model results? And How? I preassume that with a precious number size distribution of MBPAs and bacteria, the model results will be more close to reality.

The assumed diameter of MBPAs has an impact on the calculated number of MBPAs in the ocean surface layer, as we need this information to convert the DOC based estimate for polymer mass into a number concentration of particles. In contrast, the number concentration of bacteria is derived using eq. 1, so that no size information is needed for this step of the calculation. Additionally, the size of bacteria and MBPAs is relevant for the entrainment process, with the sea spray aerosol diameter imposing physical limits for the number particles that can be entrained. In our study, due to the low particle concentrations in sea water, these limits are not exceeded. The particle size is also relevant for determining the aerosol surface area used to estimate INP concentrations.

The focus on a 'mono-disperse' size distribution for bacteria and MBPAs is a simplification which allowed us to better demonstrate the impact of different factors investigated in our scenario-based approach. Also, to our knowledge there are currently no observations for size distributions of marine biological particles acting as INPs. However, it would be interesting to include realistic size distributions in future studies, because assumptions regarding the particle size may introduce substantial uncertainties, particularly because of eq. 3 which scales quadratically with particle size.

We have added the following sentences to the manuscript (l. 287):

Assuming a monodisperse size distribution allows us to better demonstrate the impact of different factors investigated in our scenario-based approach which will be described in the following paragraphs. Also, to our knowledge there are currently no observations for size distributions of individual marine species acting as INPs. However, note that the particle size may introduce substantial uncertainties, particularly because of eq. 3 which scales quadratically with particle size.

3. I think it needs more discussion concerning the model results (Fig. 6-9). For example, (1) the EF and ns linearly correlated with model INP number concentration, but the high concentration (90th) of bacteria and MABPs are not.

We are not completely sure whether we understand this comment. However, we would like to point out that the distributions of the number of bacteria and MBPAs differ across different locations and seasons, which means that a shift from the 50$^{th}$-percentile to the 90$^{th}$-percentile will affect the simulated INP concentrations accordingly (i.e. this shift might differ across regions and seasons).

(2) Which factor is more important to control the model results? It seems the high bacteria and MBPAs concentration is less important than EF and ns numbers.

This is a very valid observation and we have added a sentence highlighting this point (l. 381 ff):

Our results show that compared to the variation in INP concentrations in sea water, the enrichment factor as well as the particle type dependent ice nucleation activity have a larger impact on the estimated jet drop INP concentrations.

(3) In the MARCUS site, field INP number concentration in DJF is higher than SON, the model results also follow the same trend.

There is huge variability in the observed INP concentrations, and therefore a direct point-by-point comparison is challenging. Additionally, considering the uncertainties associated both with our simulations and the measurements, it seems difficult to pinpoint any major differences or similarities between the two MARCUS cases.

(4) Why model results in the MARCUS show larger variation, but in the MAGIC and MHD show small variation? But the field results in all sites show comparable variation.

The variability in simulated jet drop concentrations which drives the variability in simulated INPs is slightly more pronounced for the two MARCUS cases (which are also looking at a bigger domain than the MHD and the MAGIC case). We have added a corresponding sentence in l. 383ff:

The relative variability in simulated INP concentrations – which is driven by the variability in jet drop concentrations across each domain – is largest for the two MARCUS cases.

4. A suggestion: It would be nice to show the simulated INP concentration from bacteria and MBPAs on a global map and compare it to previous studies, such as Burrow et al., 2013.

[Figure]

Simulated, seasonally averaged film drop INP concentrations are lower than observed INP concentrations, with concentrations typically between $10^{-2}$ and $10^{-4}$ m$^{-3}$. The contribution of film drop INPs is based on the concentration and surface area of smaller sea spray particles ($d_{wet}$<1µm). We assume an average ice nucleation activity at 253 K of $n_s = 10^5$ m$^{-2}$ (McCluskey et al., 2018). We have also added this comparison to the manuscript (l. 395ff):

Note that simulated film drop INP concentrations based on an average ice nucleation activity at 253 K of $n_s$=$10^5$ m$^{-2}$ (McCluskey et al., 2018c) generally range between $10^{-4}$ and $10^{-2}$ m$^{-3}$ (data not shown) which is substantially lower than the observed INP concentrations.

5. Why you only compare the episodic INPs, i.e., the highest 75th percentile INPs? If you compare the median of all INP number concentrations with your model, the results might be comparable. As the paper is well structured, I just need clarification in the response.

Our initial motivation for this study was indeed the occurrence of episodically-high INP concentrations (as represented by the 75$^{th}$ percentile, i.e. the tail of the distribution). However, in Figs. 6-9 we compare

all observed values against the distribution of all simulated values for the relevant spatial domain, and only indicate the threshold for high INP values (grey line).

**Minor comments:**

1. Lines 99-102: I am not clear why you have this paragraph in the abstract. The logic flow seems been interrupted by this paragraph. Please clarify it.
With this paragraph we wanted to highlight that the presence of different ice nucleating entities may be interlinked. However, because this aspect is mentioned briefly in the previous paragraph, we have opted to remove the paragraph in question.

2. Lines 116-120: It worth mentioning that the enrichment factor is also related to particle sizes. See Fig. 5 in Quinn et al., 2015.
We have added a sentence to the manuscript to account for this aspect (l. 141ff):
Note that EF is typically given in relation to the sea salt content, and also depends on the organic/biological species in question as well as the SSA particle size (Quinn et al., 2015).

3. In Fig. 6-9, Error bars of field measurements (blue dots) are needed. I also suggest combining Fig. 6 into one figure, with (a) to (d) panels.
Error bars had been added as part of the initial technical corrections before the discussion phase, but we have increased the line thickness to make them more visible. We have only added representative error bars for the lowest INP concentrations, in order not to overcrowd our plots. We would like to maintain the four panels, as we feel that otherwise the readability (e.g., axis labels) might get compromised.

**References**

Andersen, K. H., Berge, T., Gonçalves, R. J., Hartvig, M., Heuschele, J., Hylander, S., Jacobsen, N. S., Lindemann, C., Martens, E. A., Neuheimer, A. B., Olsson, K., Palacz, A., Prowe, A. E. F., Sainmont, J., Traving, S. J., Visser, A. W., Wadhwa, N. and Kiørboe, T.: Characteristic Sizes of Life in the Oceans, from Bacteria to Whales, Ann. Rev. Mar. Sci., 8, 217–241, 2016.

Beall, C. M., Michaud, J. M., Fish, M. A., Dinasquet, J., Cornwell, G. C., Stokes, M. D., Burkart, M. D., Hill, T. C., DeMott, P. J. and Prather, K. A.: Cultivable halotolerant ice-nucleating bacteria and fungi in coastal precipitation, Atmos. Chem. Phys., 21(11), 9031–9045, 2021.

Ickes, L., Porter, G. C. E., Wagner, R., Adams, M. P., Bierbauer, S., Bertram, A. K., Bilde, M., Christiansen, S., Ekman, A. M. L., Gorokhova, E., Höhler, K., Kiselev, A. A., Leck, C., Möhler, O., Murray, B. J., Schiebel, T., Ullrich, R. and Salter, M. E.: The ice-nucleating activity of Arctic sea surface microlayer samples and marine algal cultures, Atmos. Chem. Phys., 20(18), 11089–11117, 2020.

McCluskey, C. S., Hill, T. C. J., Malfatti, F., Sultana, C. M., Lee, C., Santander, M. V., Beall, C. M., Moore, K. A., Cornwell, G. C., Collins, D. B., Prather, K. A., Jayarathne, T., Stone, E. A., Azam, F., Kreidenweis, S. M. and DeMott, P. J.: A Dynamic Link between Ice Nucleating Particles Released in Nascent Sea Spray Aerosol and Oceanic Biological Activity during Two Mesocosm Experiments, J. Atmos. Sci., 74(1), 151–166, 2017.

McCluskey, C. S., DeMott, P. J., Ma, P. -L and Burrows, S. M.: Numerical Representations of Marine Ice-Nucleating Particles in Remote Marine Environments Evaluated Against Observations, Geophys. Res. Lett., doi:10.1029/2018GL081861, 2019.

Modini, R. L., Russell, L. M., Deane, G. B., and Stokes, M. D.: Effect of soluble surfactant on bubble persistence and bubble-produced aerosol particles, J. Geophys. Res.-Atmos., 10, 1388–1400, 2013.

Orellana, M. V., Matrai, P. A., Leck, C., Rauschenberg, C. D., Lee, A. M. and Coz, E.: Marine microgels as a source of cloud condensation nuclei in the high Arctic, Proc. Natl. Acad. Sci. U. S. A., 108(33), 13612–13617, 2011.

Quinn, P. K., Collins, D. B., Grassian, V. H., Prather, K. A. and Bates, T. S.: Chemistry and related properties of freshly emitted sea spray aerosol, Chem. Rev., 115(10), 4383–4399, 2015.

Vali, G.: Quantitative Evaluation of Experimental Results an the Heterogeneous Freezing Nucleation of Supercooled Liquids, J. Atmos. Sci., 28(3), 402–409, 1971.

Wilbourn, E. K., Thornton, D. C. O., Ott, C., Graff, J., Quinn, P. K., Bates, T. S., Betha, R., Russell, L. M., Behrenfeld, M. J. and Brooks, S. D.: Ice Nucleation by Marine Aerosols Over the North Atlantic Ocean in Late Spring, J. Geophys. Res. D: Atmos., 125(4), 457, 2020.